# Smaller preferred interpersonal distance for joint versus parallel action

Laura Schmitz [1,2☯], Arran T. Reader [3☯] *

**1** Department of Neurology, University Medical Center Hamburg-Eppendorf, Hamburg, Germany, **2** Institute of Sports Science, Leibniz University Hannover, Hannover, Germany, **3** Department of Psychology, Faculty of Natural Sciences, University of Stirling, Stirling, United Kingdom

☯ These authors contributed equally to this work.
* arran.reader@stir.ac.uk

**Data Availability Statement:** Data can be publicly accessed at https://doi.org/10.17605/OSF.IO/XFV4E.

**Funding:** The authors received no specific funding for this work.

## Abstract

During social interaction, humans prefer to keep a certain distance between themselves and other individuals. This preferred 'interpersonal distance' (IPD) is known to be sensitive to social context, and in the present study we aimed to further investigate the extent to which IPD is affected by the specific type of social interaction. In particular, we focused on the contrast between joint actions, where two or more individuals coordinate their actions in space and time to achieve a shared goal, and parallel actions, where individuals act alongside each other but individually. We predicted that joint action would be associated with a smaller preferred IPD compared to parallel action. Additionally, given that this research took place in the midst of the COVID-19 pandemic, we aimed to assess whether IPD preferences are affected by individuals' concerns about infection in general, as well as COVID-19 in particular. We predicted that higher individual concerns would be associated with greater preferred IPD. To test these hypotheses, we asked participants to imagine different social scenarios (involving either joint or parallel actions alongside a stranger) and indicate, on a visual scale, their preferred IPD. The results of two experiments (n = 211, n = 212) showed that participants preferred a shorter distance when they imagined acting jointly compared to when they imagined acting in parallel. Moreover, participants who reported higher discomfort for potential pathogen contact and who were more aware of the COVID-19 context in which the study took place preferred a larger IPD in general. Our results provide further evidence that different types of social interaction shape IPD preference. We discuss potential reasons for this phenomenon and highlight remaining questions for future research.

## Introduction

During social interaction, humans prefer to keep a certain distance between themselves and other individuals. This so-called 'interpersonal distance' (IPD; [1]), which typically extends slightly beyond a person's reach, is closely monitored and constantly adjusted to remain appropriate for comfortable interaction. Crucially, the IPD that individuals prefer must be small enough for the particular interaction to be carried out smoothly, yet large enough as not

**Competing interests:** The authors have declared that no competing interests exist.

to induce discomfort ([2–4]; also see [5]). In this sense, IPD can be considered a key component of natural social interaction [1, 6–8].

Not surprisingly, IPD has been shown to be sensitive to social context: it is comparatively larger in threatening compared to safe situations [1, 3, 9–11], and in interactions with strangers compared to friends/acquaintances [12–14] and with non-human-like agents compared to human-like agents [15, 16]. Individuals also prefer to keep a greater distance if the counterpart shows an angry facial expression compared to a happy or neutral one [17–19]. Moreover, there is initial evidence suggesting that IPD depends on whether an interaction is cooperative or competitive in nature ([20]; also see [8]).

In the present study, we aimed to further investigate to what extent IPD is affected by the type of social interaction. In particular, we focused on the contrast between *joint actions*, where two or more individuals coordinate their actions in space and time to achieve a joint goal (see [21–23]), and *parallel actions*, where individuals act alongside each other but individually (see [24–26]). Both joint and parallel actions are typically (though not always) performed within a shared physical environment, and often with small distances between individuals. Imagine, for instance, volunteers from a local community fixing recent storm damage by removing a fallen tree from a pathway. Two volunteers are operating a two-man saw (each holding one end of the saw) to cut a branch of the tree in half. Two other volunteers each operate their own saw to cut different branches of the tree. Comparing these two scenarios, we would consider the former two volunteers to be acting jointly and the latter two volunteers to be acting in parallel. Note that these two scenarios are merely meant to serve as intuitive examples illustrating the distinction between joint and parallel action–they are not meant to provide a complete definition. Instead, in line with a recent, comprehensive account [24], we assume that what differentiates a joint from a parallel action is that when individuals act jointly, their actions are collectively directed towards the same goal, such as cutting the same branch (e.g., [27]; on collective/joint/shared goals, see [24, 28, 29]). Here, we asked whether the distinction between joint and parallel action might also be reflected in the distance individuals prefer to keep from one another during interaction. Understanding how personal space preferences change in these common forms of social behaviour can provide a more detailed picture of typical human interactions.

We hypothesised that individuals will prefer smaller IPD when acting jointly compared to in parallel for a number of reasons. First, previous research has shown that individuals prefer smaller IPD during cooperative compared to competitive actions ([20]; also see [8]). Given this finding, one could thus also expect that smaller IPD is preferred in joint action compared to parallel action–since joint action, in contrast to parallel action, is intrinsically cooperative in nature. Second, acting at closer distances could be preferred because it potentially facilitates interpersonal coordination. This is because successful coordination often requires being able to infer others' action goals and intentions by observing their actions (e.g., [21, 30, 31]). Thus, to facilitate action observation, and thereby interpersonal coordination, individuals might prefer to act more closely to their interaction partners. Third, during joint action, individuals might experience a sense of joint agency, increasing their feeling of 'togetherness' and 'integration' with the other agent (for recent reviews on joint agency, see [32, 33]), which in turn could lead to a preference for smaller IPD.

Further support for our hypothesis comes from a recent study showing that IPD is linked to the subjective quality of an interaction, such that smaller IPD correlates with higher enjoyment [34]. One reason for this link could be that interpersonal rapport is often established and expressed through non-verbal means such as facial expressions (e.g., [35–38]; also see [39]) and these are easier to read from a closer distance. Consequently, we reasoned that individuals

might prefer closer distances during joint (compared to parallel) action in order to establish good social rapport with the other agent (although we do not specifically test this).

As the present study was conducted in early 2022, during the COVID-19 pandemic, we considered that COVID-related concerns might naturally have an effect on participants' IPD preferences. After all, research on this topic showed that people preferred to keep farther apart from other individuals during the pandemic [40, 41], in line with social distancing guidelines for the avoidance of virus transmission. It was also shown that preferred IPD is sensitive to the wearing of face masks [18, 42–44] and to others' infection status [45]. Moreover, changes in distance preferences have been found to be associated with individual differences in anxiety [42, 46] and in perceived risk of infection [40, 46]. In light of these findings, we predicted that participants who are generally more concerned about infection (i.e., view themselves as more susceptible) and consider the COVID-19 context when judging preferred distance will prefer greater IPD. In addition, we were also interested in whether the latter two factors might influence the degree to which individuals' preferred IPD differs between joint and parallel action. That is, whether these factors might have a greater effect on one type of interaction compared to the other.

In sum, the aim of the present study was to test whether the interpersonal distance (IPD) people prefer to keep from others is influenced by the type of social interaction they are involved in, i.e., whether they are acting jointly [22] or in parallel [24] with others. We tested our hypotheses in two experiments, with the second performed to control for a possible confound in the first. This confound was related to the use of shared objects in joint action scenarios (see Experiment 1 discussion). We predicted that joint action will be associated with a smaller preferred IPD compared to parallel action (H1). Additionally, based on recent research during the COVID-19 pandemic, we aimed to assess whether IPD preferences might be affected by individuals' concerns about infection in general and about COVID-19 in particular. We predicted that higher individual concerns will be associated with greater preferred IPD (H2).

## Experiment 1

To test H1, we presented participants with descriptions of social scenarios which involved interacting with another (unfamiliar) person. For each scenario, participants were asked to indicate, on a visual scale, the smallest distance from the other person at which they would feel comfortable (see Fig 1). The scenarios came in two versions that differed with respect to the type of social interaction (joint versus parallel) described, but were comparable with respect to the overall action context; see Table 1. As such, we used a repeated measures design to examine the influence of social context (joint action, parallel action) on preferred IPD. Additionally, to test H2, we assessed participants' perceived vulnerability to disease via a questionnaire and asked them to report their subjective awareness of COVID-19 at the moment of the study. See Experiment 1 Materials below for details.

### Sample size estimation

Our sample size was based on an a priori power analysis (alpha = 0.05, power = 0.80, two-tailed paired $t$-test) which showed that 200 participants would be sufficient to detect a small effect (Cohen's $d$ = 0.2), i.e., we would be able to detect a potentially small difference in IPD between joint and parallel actions. In case we had to exclude any participants during analysis due to technical problems or other reasons (see data exclusion criteria in Experiment 1 'Data analysis' section), we decided to collect a sample of 220 individuals.

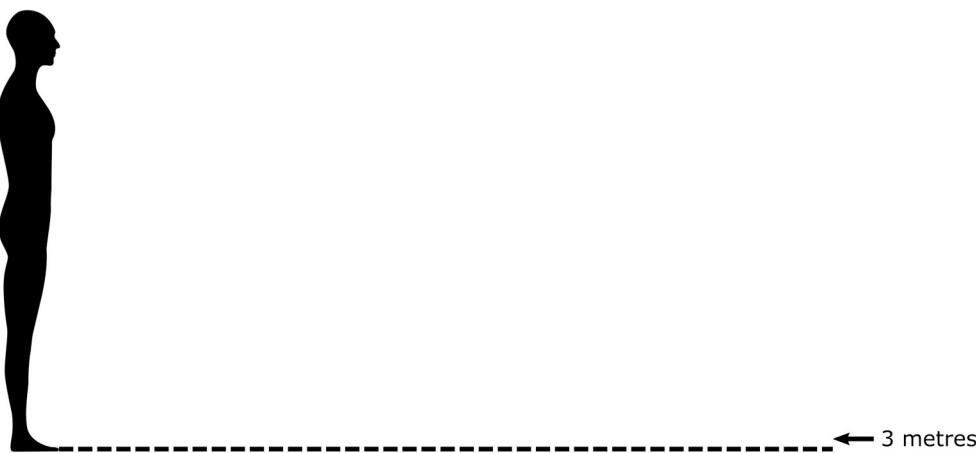

**Fig 1. IPD rating scale.** Participants were asked to indicate the smallest distance from the depicted person at which they would be comfortable, by clicking on the image.

## Participants

We tested a sample of participants from two Western European countries (Germany and the UK) with comparable 'baseline' IPD preferences [14]. We recruited 110 participants from Germany and 110 participants from the UK via Prolific (https://www.prolific.co/). All participants were required to be fluent in English, aged between 25 and 60 years old, and to have lived in the UK or Germany for the majority of the past two years at the time of testing. Participants were also required to have a 100% approval rating on Prolific. We collected a balanced sample of 50% male and 50% female participants.

Following exclusions (see Experiment 1 'Data analysis' section), the sample used for data analysis consisted of 211 individuals aged between 25 and 60 years old, mean±SD age = 35.8 ±9.21 years (106 women, 105 men). All participants provided informed consent and the experiment was approved by the University of Stirling General University Ethics Panel (review reference: GUEP 2021 4316 3574). Participants received monetary compensation for their time (£1.25, equivalent to £7.50 per hour). Authors did not have access to information that could identify individual participants during or after data collection

## Materials

The study was conducted as an online survey using Qualtrics (https://www.qualtrics.com). In the survey, participants were presented with descriptions of social scenarios which involved interacting with another (unfamiliar) person. For each scenario, participants were asked to indicate the smallest distance from the other person at which they would be comfortable, on a visual scale from 0 to 3 metres (see Fig 1). The scale was similar to the Interpersonal Visual Analog Scale [47] and the Pedersen personal space scale [48]. Previous research has shown that effects observed using these types of rating scales tend to align well with effects observed using more naturalistic methods (47). Our scale pictured a gender-neutral human silhouette with a relative height of 1.7 m. The image was 996 x 472 pixels in size (600 dpi), scaled to individual monitor size, and could be clicked on using the computer mouse. Thus, using the mouse as a response device, participants could easily indicate their smallest preferred IPD by clicking on (or above) the scale at a distance they judged as appropriate. Participants could provide a minimum distance of 0 metres and a maximum distance of 3 metres. Henceforth, we refer to this task as the 'IPD rating task'.

**Table 1. Text (exact wording) presented to participants for each scenario in joint and parallel versions.**

| Experiment | Scenario and opening | Version | |
|---|---|---|---|
| | | Joint | Parallel |
| 1 | Storm Damage<br>Imagine that you are working with members of your local community to fix recent storm damage. | A tree has fallen over a pathway, and **you are using a 'two-man saw' to cut one of the branches** in half. **You are holding one end of the saw** and **an unfamiliar member of your community** is standing opposite to you and **is holding the other end of the saw.** | A tree has fallen over a pathway, and **you are using a saw to cut one of the branches** in half. **An unfamiliar member of your community** is standing opposite to you and **is using a different saw to cut another branch** in half. |
| | Exercise Class<br>Imagine that you are at an exercise class. | **You and an unfamiliar member of the class** are standing opposite each other and **throwing a ball back and forth.** | **You and an unfamiliar member of the class** are standing opposite each other. Each of you is **lifting a ball up and down.** |
| | Office Work | Imagine that you are moving a rectangular table at your place of work. **You are lifting one end of the table while a new, unfamiliar colleague is lifting the opposite end.** | Imagine that you are sitting at a rectangular table at your place of work. **You are sitting at one end of the table** and working on a laptop **while a new, unfamiliar colleague is sitting at the opposite end** and is working on their laptop. |
| | Cafeteria<br>Imagine that you are volunteering at a cafeteria, preparing plates of food for an event in your local community. | **You are placing rice on plates** and then putting the plates on a conveyor belt. **An unfamiliar member of your community** is standing opposite to you and **is adding vegetables to these plates.** | **You are placing rice and vegetables on plates** and then putting the plates on a conveyor belt. **An unfamiliar member of your community** is standing opposite to you and is **placing rice and vegetables on other plates.** |
| | Event Preparation<br>Imagine that you are preparing banners for an event in your local community. | **You are** standing at a table **writing text on a banner. An unfamiliar member of your community** is standing opposite to you and **is adding illustrations to the same banner.** | **You are** standing at a table **writing text on a banner and adding illustrations. An unfamiliar member of your community** is standing opposite to you and is **preparing another banner.** |
| 2 | Storm Damage<br>Imagine that you are working with members of your local community to fix recent storm damage. | A large tree has fallen over a pathway, and **you are using a saw to cut a branch into smaller pieces so that the tree can be moved. An unfamiliar member of your community** is standing opposite to you and **cutting another part of the same branch.** | A large tree has fallen over a pathway, and **you are using a saw to cut a branch into smaller pieces so that it can be moved. An unfamiliar member of your community** is standing opposite to you and **cutting a different branch.** |
| | Exercise Class<br>Imagine that you are at an exercise class. | **You are performing a step routine** while **an unfamiliar member of the class is performing the same routine** opposite to you. The class instructor asks you to **perform your steps in synchrony.** | **You are performing a step routine** while **an unfamiliar member of the class is performing a different routine** opposite to you. The class instructor asks you to **perform your routines one after the other**, so she can check whether you perform all movements correctly. |
| | Office Work<br>Imagine that you are sitting at a rectangular table at your place of work. **You are sitting at one end of the table** and working on a laptop while a **new, unfamiliar colleague is sitting at the opposite end** and is working on their laptop. | You are working quietly with headphones on to fill in **the same Excel spreadsheet** before a meeting that you will both attend later in the day. | You are working quietly with headphones on to fill in **separate Excel spreadsheets** before separate meetings later in the day. |
| | Cafeteria<br>Imagine that you are volunteering at a cafeteria, cooking food for an event in your local community. | **You are standing at a stove preparing one component for a meal** while **an unfamiliar member of your community** is standing opposite to you at a different stove, **preparing the other component of the meal.** | **You are standing at a stove preparing a meal** while **an unfamiliar member of your community** is standing opposite to you at a different stove, **preparing a different meal.** |
| | Drumming Lesson<br>Imagine that you are learning to play bongo drums. **You are sharing the lesson with an unfamiliar student**, who is sitting opposite. | Your teacher asks you to **play at the same time as the other student, whilst matching your rhythms.** | Your teacher asks you to **play one at a time, so he can hear your progress.** |

The scenarios described joint and parallel actions, such that there were always two matched scenario versions which differed only regarding the joint-parallel distinction while being comparable in other respects. There were five different scenarios and each scenario was presented in a joint and parallel version, giving us 10 scenarios in total. To illustrate, one of the scenarios

would take place at an exercise class, where participants were asked to imagine passing a ball back and forth with another agent (joint), or lifting a ball up and down, in parallel to another agent doing the same (parallel). Another scenario would involve cutting branches of wood, either together with another agent, using a 'two-man saw' (joint), or in parallel to another agent, each using their own saw (parallel). All scenarios are displayed in Table 1.

To assess concerns about infection risk (given the global COVID-19 pandemic), we used the Perceived Vulnerability to Disease (PVD) self-report scale [49], which measures perceived vulnerability along the two dimensions Perceived Infectability (beliefs about susceptibility to infectious disease) and Germ Aversion (emotional discomfort in scenarios with the potential for pathogen transmission). The PVD scale features statements to which participants are asked to provide their level of agreement, such as 'I have a history of susceptibility to infectious disease' (Perceived Infectability) and 'I prefer to wash my hands pretty soon after shaking someone's hand' (Germ Aversion); see S1 Table for a complete list of the PVD items. Responses were provided on a scale from 1 (Strongly disagree) to 7 (Strongly agree).

To measure whether participants were thinking of COVID-19 while completing the task, we presented participants with a 'COVID-context awareness' question at the very end of the survey. Participants were asked to indicate how much they agreed with the following statement: 'While I was reporting the smallest distance I would feel comfortable being near another person in the previous scenarios, I was thinking about COVID-19 containment measures (e.g., social distancing, mask wearing, infection testing).' Participants indicated their agreement on a scale from 1 (Strongly disagree) to 7 (Strongly agree).

## Procedure

Participants were tested on Monday 14th February 2022. At the end of the previous week the cumulative COVID-19 cases per 100,000 people were approximately 15,000 in Germany and 27,000 in the UK. 77% of individuals in Germany and 78% in the UK had received at least one COVID-19 vaccination (https://ourworldindata.org/coronavirus). Participants took part using their own computers.

Participants were first informed that they would be presented with a series of scenarios to imagine, each of which containing an encounter between them and a person they are unfamiliar with, and that they would have to report the smallest distance they would feel comfortable being near the other person. Participants were requested to read the scenario descriptions carefully and it was emphasised that all of the scenarios are different, even if the context is similar. The ten scenarios were then presented in a randomised order. For each scenario, participants reported the smallest preferred IPD on a scale from 0 to 3 metres (see Fig 1). Participants were asked to assume that the action can be effectively performed at any distance.

Following the IPD rating task, participants were asked to report their age in years, their gender (man, woman, non-binary), and the country in which they had spent the majority of the time living over the past two years. Finally, they were presented with the PVD scale (with items in a randomised order) and the COVID-context awareness question (see Experiment 1 'Materials' section). On average, participants needed approximately eight minutes to complete the entire survey.

## Data analysis

Participants were excluded from analysis if they did not finish the survey, spent less than 3 minutes (n = 2) or more than 30 minutes (n = 2) on the survey, did not spend the majority of the last two years in the country from which they were recruited (n = 5), or provided an IPD rating of less than 5 cm as a mean across all scenarios.

Responses to the IPD rating task were converted from pixels into metres. The smallest possible value that participants could provide when clicking on the image was 71 pixels in the $x$ dimension, and the size of the image was equivalent to 258 pixels per metre. As such, preferred IPD in metres was calculated by using the following formula: (pixel value—71)/258.

As dependent measures, we calculated, per participant, the mean value for preferred IPD in the five joint scenarios (*jointIPD*) and five parallel scenarios (*parallelIPD*), as well as the grand mean across all ten scenarios (*meanIPD*). We also calculated the difference in preferred IPD between matched joint and parallel versions (parallel—joint), such that a larger value indicates a preference for a closer distance in joint action. The mean of these difference values was calculated per participant (*diffIPD*). The two dimensions of the PVD scale (Perceived Infectability and Germ Aversion) were computed by taking the mean of the relevant items (reverse coded where necessary).

To test H1 (predicting greater IPD for joint compared to parallel action), we used a two-tailed paired samples *t*-test to compare participants' *jointIPD* to *parallelIPD*, after using a Shapiro-Wilk test to confirm normality. To test H2 (predicting a general influence of individual concerns about infection and COVID-19 on IPD), we ran a multiple linear regression using *meanIPD* as a dependent variable. The independent variables (i.e., covariates) were Country (dummy coded, to control for any possible regional differences), Perceived Infectability, Germ Aversion, and COVID-context awareness. Regression was performed using the 'Enter' method, and the suitability of the analysis was confirmed by checking for the linearity of the relationship between independent and dependent variables, the absence of outliers (standardised residual > 3), the independence of observations, homoscedasticity, normal distribution of residuals, absence of multicollinearity. Standardised residuals > 3 were removed (only once per regression analysis reported).

For the regression, we specifically predicted that Perceived Infectability and COVID-context awareness would positively predict preferred IPD. As Perceived Infectability provides an index of individuals' subjective beliefs about their susceptibility to catch an infectious disease from other people, this should be particularly relevant during social interaction. The second dimension of the PVD scale, namely Germ Aversion, was considered slightly less relevant because it focuses more on individuals' concerns about potential pathogen transmission from objects previously touched by other people rather than on direct interaction. However, a study conducted during the early phase of the COVID-19 pandemic [40] indicated that people with greater fear of being contaminated by pathogens (as indicated by the Germ Aversion dimension of the PVD scale) showed greater COVID-related changes in peripersonal space, which is a different, but potentially related concept to IPD [6] (but see [50]). Thus, Germ Aversion might also predict preferred IPD. A further study, that we were not aware of at the time of pre-registration, also found a link between Germ Aversion and IPD preferences [51].

In addition to our hypothesis tests, we ran two exploratory analyses. First, to evaluate whether participants' concerns about infection in general and about COVID-19 in particular might influence the degree to which preferred IPD differs between joint and parallel action, we ran another multiple linear regression, this time using *diffIPD* as a dependent variable. Second, to examine whether differences in preferred IPD were consistent across our five scenarios, we computed the mean IPD difference between joint and parallel action separately for each scenario. We then tested whether the difference in each scenario was significantly different from zero by using one sample Wilcoxon signed-rank tests (due to deviations from normality). If the difference was greater than 0, this meant that participants, on average, preferred a smaller distance during joint compared to parallel action; if it was smaller than 0, the reverse was true. If particular scenarios differed from the others in this respect (e.g., showing no difference while the others showed a difference), these scenarios were then compared to the others by

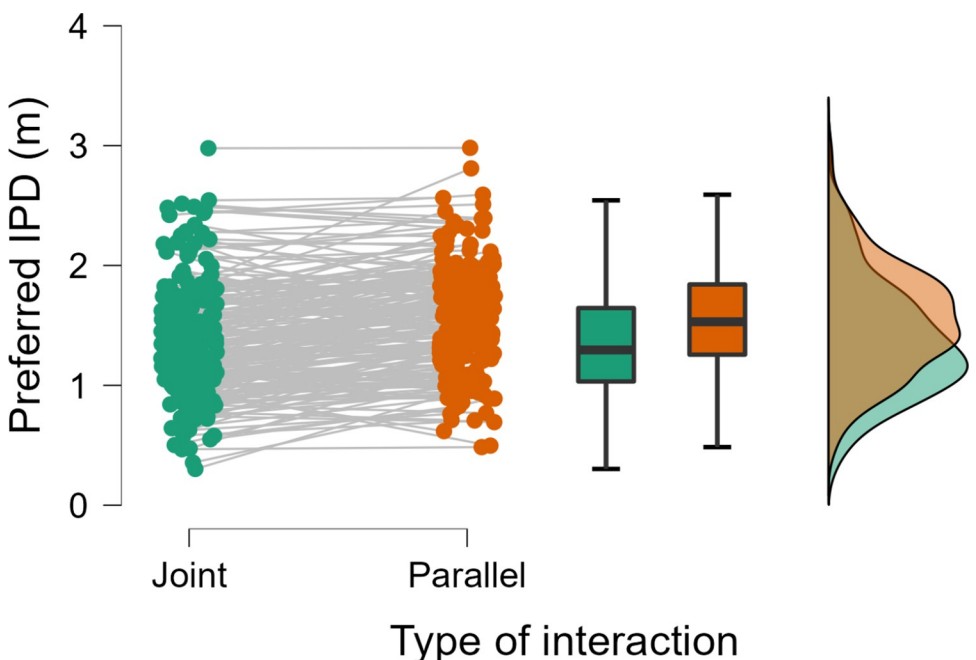

**Fig 2. Individual responses, boxplots, and distributions for preferred IPD (Experiment 1).**

using linear mixed models. Note that the details of this second exploratory analysis were not preregistered.

All analyses were performed with JASP, version 0.16 [52] or R, version 4.0.5 [53].

## Results

**Main analyses.** Consistent with H1, we observed that participants displayed a preference for a smaller IPD in the joint action scenarios (mean±SE = 1.36±0.0331 m) compared to the parallel action scenarios (1.55±0.0309 m), $t(210) = -9.26$, $p < .001$, Cohen's d = -0.637, 95% CI = [-0.785, -0.489] (Fig 2), with 77% of participants displaying an effect in this direction.

In the multiple regression analysis with *meanIPD* as the dependent variable, a statistically significant model was observed, $F(4, 204) = 7.36$, $p < .001$, adj. $R^2 = .109$. Consistent with H2, the two covariates Germ Aversion and COVID-context awareness, yet not the other two (i.e., Perceived Infectability and Country), were found to be statistically significant predictors of IPD preference (Table 2).

**Exploratory analyses.** The multiple regression analysis with *diffIPD* as the dependent variable did not result in a statistically significant model, $F(4, 204) = 0.905$, $p = .462$. The full output is reported in S2 Table.

**Table 2. Multiple regression analysis for *meanIPD* (Experiment 1).** Statistically significant predictors displayed in bold.

|  | Unstandardised estimate | Standard error | β | t | p |
|---|---|---|---|---|---|
| Intercept | 0.915 | 0.120 |  | 7.63 | < .001 |
| Country | -0.000465 | 0.0556 | -.000551 | -0.00837 | .993 |
| Perceived Infectability | -0.00503 | 0.0251 | -.0141 | -0.200 | .841 |
| **Germ Aversion** | 0.106 | 0.0294 | .263 | 3.60 | .000399 |
| **COVID-context awareness** | 0.0372 | 0.0142 | .179 | 2.62 | .00939 |

To better understand this null result we ran a post hoc Bayesian multiple linear regression in JASP. We used the default JZS prior distribution (r scale of 0.354) for coefficients, and a uniform model prior which characterises all possible models as equally likely. The null model (containing only the intercept) was found to be the most probable given the data, with P (model|data) = .381. The second most probable model contained only COVID-context awareness as a predictor, with P(model|data) = .204. However, the Bayes factor for this model suggested that the observed data were 1.87 times more likely under the null model ($BF_{10}$ = 0.535). For all other models, P(model|data) < .1, and the observed data were at least 6.3 times more likely under the null model in each case (i.e., $BF_{10} \leq 0.159$).

The mean IPD difference between joint and parallel action, computed separately for each scenario, showed that differences across scenarios ranged from 0 to 0.35 m (Storm Damage: 0.347 m, Exercise Class: -0.000276 m, Office Work: 0.301 m, Cafeteria: 0.0827 m, Event Preparation: 0.219 m). The mean differences for all scenarios were found to be significantly different from zero, as indicated by one sample Wilcoxon signed-rank tests ($p \leq .00126$), except for Exercise Class ($p$ = .925).

Consequently, a linear mixed model was used to test whether Exercise Class differed from the other scenarios. The model included the factor Scenario as a fixed effect, for which we coded Exercise Class as a reference group. Moreover, we included random intercepts for each participant in the model. We decided against adding random slopes for Scenario as the model did not converge. The model coefficients were tested for significance using Wald $t$-tests. We found that the Exercise Class scenario differed significantly from the Event Preparation scenario (B = 0.22, $t(844)$ = 4.38, $p < .001$), the Storm scenario (B = 0.35, $t(844)$ = 6.94, $p < .001$), and the Office scenario (B = 0.30, $t(844)$ = 6.01, $p < .001$). Relative to the Cafeteria scenario, there was no significant difference (B = 0.08, $t(844)$ = 1.66, $p$ = .098).

## Discussion

Our results showed that participants, on average, preferred to keep a shorter distance between themselves and another (unfamiliar) individual when they imagined acting jointly with that individual (joint = 1.36 m) compared to when they imagined acting in parallel (parallel = 1.55 m). This difference in preferred IPD for joint versus parallel action (mean difference = 0.19 m) was not affected by participants' individual concerns about infection in general and about COVID-19 in particular. However, participants' overall IPD preference (averaged across joint and parallel action) was modulated by participants' discomfort in scenarios with the potential for pathogen transmission (as measured by the dimension 'Germ Aversion' of the PVD scale; [49]). In addition, it was modulated by participants' momentary concern about COVID-19. As such, participants who reported higher discomfort for potential pathogen contact and who were more aware of the COVID-19 context preferred to keep further away from a co-acting individual.

Finally, we noticed that the difference in preferred IPD for joint versus parallel action was observed consistently across four of our five scenarios. Only in the Exercise Class scenario, no such difference was observed. The joint version of this scenario differed from all others in one noticeable aspect: whereas in all other scenarios, two individuals were concurrently acting upon a shared object (i.e., operating a two-man saw, lifting a table, placing objects on a conveyor belt, writing/drawing on a banner), the individuals in the Exercise Class scenario were passing a ball back and forth. Thus, individuals in the latter scenario were not as constrained by the shared object (because the ball was not acted upon at the same time). This opens the possibility that the differences in preferred IPD between joint and parallel action, present in all but the Exercise Class scenario, might be due to the fact that individuals in the joint version

were constrained by the shared object. In the parallel version, there was no such constraint in any of the scenarios because individuals acted upon separate objects.

Note that, to avoid the above concern, participants had been explicitly instructed to assume that all actions "can be effectively performed at any distance". Despite this instruction, we cannot safely exclude that participants were not affected by the shared objects.

## Experiment 2

We conducted Experiment 2 to ensure that the differences in preferred IPD between joint and parallel action observed in Experiment 1 cannot be explained by the fact that in the particular scenarios we chose, joint actions always required the concurrent manipulation of a shared object (e.g., a two-man saw) while parallel actions required the manipulation of separate objects. Therefore, in Experiment 2, we adjusted all scenarios such that neither joint nor parallel actions required concurrent manipulation of a shared object; instead, individuals always acted upon separate (or no) objects.

### Method

Unless otherwise stated, the methods were identical to those used in Experiment 1.

### Participants

We recruited another 220 participants (110 from Germany, 110 from the UK), none of which had taken part in the previous experiment. Following exclusions (see Experiment 2 'Data analysis' section below), the sample used for data analysis consisted of 212 participants aged between 25 and 60 years old, mean±SD age = 35.4±8.84 years (108 men, 102 women, 2 non-binary individuals).

### Materials

We developed new scenarios to eliminate the potential confound discussed above. In order to still keep the scenarios as comparable to those in Experiment 1 as possible, we did not change the overall action context (except for one scenario) but only slightly adjusted the parameters of the joint/parallel actions (see Table 2).

**Procedure.** Participants were tested on Monday 11th April 2022. At the end of the previous week the cumulative COVID-19 cases per 100,000 people were approximately 27,000 in Germany and 32,000 in the UK. 77% of individuals in Germany and 79% in the UK had received at least one COVID-19 vaccination.

**Data analysis.** We excluded four participants for spending over 30 minutes on the survey and four participants for not spending the majority of the previous two years in the country from which they were recruited.

### Results

**Main analyses.** A Shapiro-Wilk test indicated a statistically significant deviation from normality for the comparison of *jointIPD* and *parallelIPD*. Thus, a Wilcoxon signed-rank test was used. Replicating the pattern from Experiment 1, we observed that participants displayed a preference for a smaller IPD in the joint action scenarios (mean±SE = 1.49±0.0333 m) compared to the parallel action scenarios (1.60±0.0321 m), W = 5855.5, p < .001, r = -.481, 95% CI = [-.592, -.354] (Fig 3), with 68% of participants displaying an effect in this direction.

In the multiple regression analysis with *meanIPD* as the dependent variable, a statistically significant model was observed, $F_{(4, 207)} = 6.51$, $p < .001$, adj. $R^2$ = .0946. As in Experiment 1,

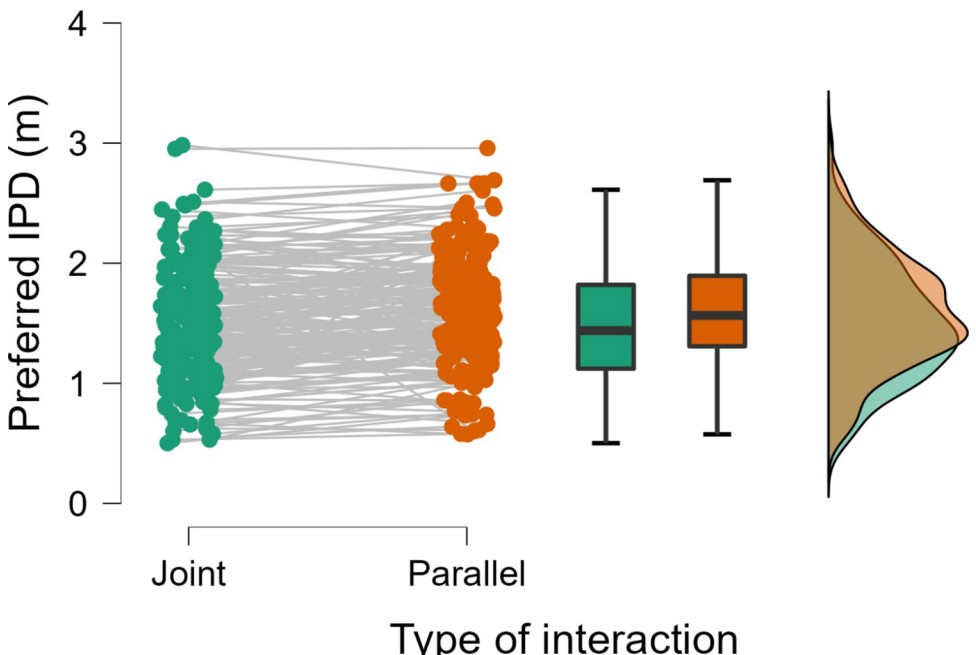

**Fig 3. Individual responses, boxplots, and distributions for preferred IPD (Experiment 2).**

Germ Aversion and COVID-context awareness were found to be statistically significant predictors of IPD preference (Table 3).

**Exploratory analyses.** The multiple regression analysis with *diffIPD* as the dependent variable did not result in a statistically significant model, $F(4, 205) = 0.841$, $p = .501$. The full output is reported in S3 Table.

As for Experiment 1, we also ran a Bayesian multiple linear regression analysis to follow up on this result, using the same parameters previously described. The null model was found to be the most probable given the data, with P(model|data) = .417. The second most probable model contained only COVID-context awareness as a predictor, with P(model|data) = .133. However, the Bayes factor for this model suggested that the observed data were 3.13 times more likely under the null model ($BF_{10} = 0.319$). For all other models, P(model|data) < .1, and the observed data were at least 4.7 times more likely under the null model in each case (i.e., $BF_{10} \leq 0.214$).

Mean differences for the scenarios ranged between 0.03 and 0.19 m (Storm Damage: 0.0255 m, Exercise class: 0.194 m, Office Work: 0.117 m, Cafeteria: 0.101 m, Drumming Lesson: 0.104 m). The mean differences for the scenarios were found to be significantly different from zero using one-sample Wilcoxon signed-rank tests ($p < .001$), except for Storm Damage ($p = .0904$).

**Table 3. Multiple regression analysis for *meanIPD* (Experiment 2).** Statistically significant predictors displayed in bold.

|  | Unstandardised estimate | Standard error | β | t | p |
|---|---|---|---|---|---|
| Intercept | 1.07 | 0.126 |  | 8.51 | < .001 |
| Country | -0.0499 | 0.0600 | -.0549 | -0.832 | .406 |
| Perceived Infectability | 0.0202 | 0.0278 | .0507 | 0.728 | .468 |
| **Germ Aversion** | 0.0742 | 0.0300 | .177 | 2.47 | .0141 |
| **COVID-context awareness** | 0.0476 | 0.0153 | .212 | 3.12 | .00209 |

To follow up on the latter, we used a linear mixed model to test whether Storm Damage differed from the other scenarios. The model included the factor Scenario as a fixed effect, for which we coded Storm Damage as a reference group. Moreover, we included random intercepts for each participant in the model. We did not add random slopes for Scenario as the model did not converge. The model coefficients were tested for significance using Wald $t$-tests. We found that the Storm Damage scenario differed significantly from the Exercise Class scenario (B = 0.17, $t(848)$ = 3.88, $p < .001$) and the Office scenario (B = 0.09, $t(848)$ = 2.13, $p = .034$). Comparisons with the Drumming Lesson scenario and the Cafeteria scenario were not statistically significant (Drumming Lesson: B = 0.08, $t(848)$ = 1.81, $p = .070$; Cafeteria: B = 0.08, $t(848)$ = 1.74, $p = .083$).

## Discussion

In Experiment 2, we replicated the main findings from Experiment 1. Participants preferred to keep a shorter distance between themselves and another (unfamiliar) individual during joint action (joint = 1.49 m) compared to parallel action (parallel = 1.60 m). As in Experiment 1, this difference in preferred IPD for joint versus parallel action (mean difference = 0.11 m) was not affected by participants' individual concerns about infection in general and about COVID-19 in particular. However, participants' overall IPD preference was modulated by participants' discomfort in scenarios with the potential for pathogen transmission and by participants' momentary concern about COVID-19, such that participants who reported higher discomfort for potential pathogen contact and higher awareness of the COVID-19 context preferred to keep further away from a co-acting individual.

Importantly, in Experiment 2, we had modified the scenarios such that neither joint nor parallel actions required concurrent manipulation of a shared object; and instead, individuals always acted upon separate (or no) objects. This was done to ensure that the differences in preferred IPD between joint and parallel action observed in Experiment were not simply caused by participants in Experiment 1 feeling distance constraints due to the shared objects involved in the joint actions. Replicating the difference in preferred IPD between joint and parallel action in Experiment 2 provides indication that this difference cannot be ascribed to different spatial constraints caused by the to-be-manipulated objects, since these constraints were exactly the same for joint and parallel actions in Experiment 2. This lets us conclude that the difference in preferred IPD we observe across experiments can be ascribed to our main manipulation, i.e., the two different types of social interaction.

In Experiment 2, the difference in preferred IPD for joint versus parallel action was observed consistently across four of our five scenarios. Only in the Storm Damage scenario, no such difference was observed. In this scenario, participants were asked to imagine that they are helping to fix storm damage. Specifically, participants were asked to imagine that a large tree has fallen over a pathway and they are using a saw to cut a branch into smaller pieces so that the tree can be moved. They were told that an unfamiliar member of the community is standing opposite them and is cutting either "another part of the same branch" (joint version) or "a different branch" (parallel version). One might argue here that in both versions of this scenario, the action feels rather 'joint' in that the co-acting individuals are acting upon the same tree with the joint goal of removing it from the pathway. The minor difference (cutting the same branch or a different branch of the tree) might have been easily overlooked or regarded as not essential with respect to the overall goal. It is possible that for this reason, individuals' IPD preferences did not differ between the joint and parallel version of the Storm Damage scenario. Alternatively, the comparable IPD ratings for the two versions could reflect the use of a potentially dangerous tool, such that individuals might maintain a more consistent distance for safety reasons, possibly overriding personal comfort.

## General discussion

The aim of the present study was to investigate whether 'interpersonal distance' (IPD; [1]), i.e., the distance individuals typically keep between themselves and others, is affected by the type of social interaction individuals are involved in. Specifically, we focused on the contrast between joint actions, where two or more individuals coordinate their actions in space and time to achieve a joint goal (e.g., [22]), and parallel actions, where individuals act alongside each other but individually (e.g., [24]). Drawing on previous research on joint action (e.g., [20, 21, 32, 34]), we predicted that joint action will be associated with a smaller preferred IPD compared to parallel action (H1). Additionally, based on research during the COVID-19 pandemic (e.g., [40, 41]), we aimed to assess whether IPD preferences might be affected by individuals' concerns about infection in general and about COVID-19 in particular. We predicted that higher individual concerns will be associated with greater preferred IPD (H2).

To test these hypotheses, we presented participants with descriptions of social scenarios which involved interacting with another (unfamiliar) person. For each scenario, participants were asked to indicate, on a visual scale, the smallest distance from the other person at which they would feel comfortable (see Fig 1). The scenarios came in two versions that differed with respect to the type of social interaction (joint versus parallel) described, but were comparable with respect to the overall action context. Additionally, we assessed participants' perceived vulnerability to disease via a questionnaire and asked them to report their subjective momentary awareness of COVID-19.

Our results of Experiment 1 showed that participants, on average, preferred to keep a shorter distance between themselves and another individual when they imagined acting jointly with that individual compared to when they imagined acting in parallel. This difference in preferred IPD for joint versus parallel action (diffIPD) was not affected by participants' individual concerns about infection in general and about COVID-19 in particular. However, participants' overall IPD preference (averaged across joint and parallel action) was modulated by participants' discomfort in scenarios with the potential for pathogen transmission (as measured by the dimension 'Germ Aversion' of the PVD scale; [49]) and by participants' momentary concern about COVID-19. That is, participants who reported higher discomfort for potential pathogen contact and who were more aware of the COVID-19 context preferred to keep further away from a co-acting individual.

One concern of Experiment 1 was that in the social scenarios we presented to participants, joint actions always required the concurrent manipulation of a shared object (e.g., a two-man saw) while parallel actions required the manipulation of separate objects. The preference for closer IPD in joint compared to parallel action could thus possibly be explained by the constraint imposed by the shared object in the joint action. To address this possibility, we conducted Experiment 2 where we adjusted all scenarios such that neither joint nor parallel actions required concurrent manipulation of a shared object; instead, individuals always acted upon separate (or no) objects.

The results from Experiment 2 replicated those from Experiment 1 (albeit with a smaller effect size). Participants preferred to keep a shorter distance between themselves and another individual during joint action compared to parallel action, and this difference in preferred IPD was not affected by participants' individual concerns about infection in general or about COVID-19 in particular. Consistent with Experiment 1, participants who reported higher discomfort for potential pathogen contact and higher awareness of the COVID-19 context generally preferred to keep further away from a co-acting individual. As, in Experiment 2, we had eliminated the potential spatial constraints in the joint scenarios that participants in Experiment 1 might have experienced, we conclude, for the time being, that the difference in

preferred IPD observed across the two experiments is not due to those constraints but can be ascribed to our main manipulation, i.e., the two different types of social interaction.

Given that our findings cannot be solely explained by the presence of a shared object, there are several alternative explanations. In the introduction we proposed possible reasons for why we might observe such effects, and these may variably apply to the different scenarios used in our experiments (in keeping with the variability in scenario-specific effect sizes). Notably, previous research has shown that individuals prefer smaller IPD during cooperative compared to competitive actions [20]. Our results are mostly in line with this finding, indicating that such a preference is not only reflective of different aims during interaction (i.e., to cooperate or compete), but perhaps also of a broader preference for closer IPD when cooperating compared to simply acting near another individual. This first explanation may be sufficient for explaining most scenarios, whether they involved joint action in the sense of interpersonal motor coordination (such as using a two-man saw), or in a broader conceptual fashion (e.g., preparing a meal, completing a shared spreadsheet).

A second explanation for our results is that acting at closer distances could be preferred because it potentially facilitates interpersonal coordination. In order to coordinate effectively, individuals must be able to infer others' action goals and intentions by observing their actions. This facilitates coordination by ensuring that changes in the partner's behaviour can be responded to rapidly and in a predictive fashion (e.g., [21, 54]). That the greatest difference in preferred IPD was observed in scenarios where close motor coordination was necessary (Experiment 1: Storm Damage, Experiment 2: Exercise Class) may provide some support for this argument. A third possible explanation is that during joint action, individuals might experience a sense of joint agency, increasing their feeling of 'togetherness' and 'integration' with the other agent [32, 33], resulting in a smaller IPD preference. In a related fashion, there is some evidence that IPD is linked to the subjective quality of an interaction, such that smaller IPD correlates with higher enjoyment [34].

Ultimately, our results only highlight that differences in IPD preference may exist for different types of social interaction. They cannot explain why these differences occur. Indeed, it remains possible that effects observed in our experiments are not related to social cognition but simply arise due to pragmatism: participants may desire to be closer in joint action scenarios to make it easier to talk to the other individual or see what they are doing. For example, improved vocal interaction may support shared task success in the Cafeteria scenario (for querying ingredients or cooking technique), and improved vision may be beneficial for the joint Exercise Class scenario in Experiment 2 (where visual feedback can facilitate performance). Similar effects might also be observed in non-social scenarios in which one can benefit from being closer to a visual or auditory stimulus. However, this explanation may not hold true in all cases. In the Drumming scenario auditory feedback could be sufficient for synchronised performance in the joint version, and the mean difference in distance of 0.104 m between conditions seems unlikely to greatly facilitate visual perception. Furthermore, in the Office Work scenario in Experiment 2, individuals are wearing headphones and there is no benefit to sitting closer in the joint version of the task. However, it is not possible to exclude that participants imagine interactions beyond those they are presented with—for example, removing their headphones to talk about the shared spreadsheet in the joint Office Work scenario, in which case improved vocal interaction may be considered.

It is also worth noting that in Experiment 2, two of the joint action scenarios (Exercise Class and Drumming Lesson) contained an additional element of coordination, namely interpersonal synchrony. This was due to the rhythmic nature of the performed actions, stepping and drumming, respectively. Since previous research has shown that synchronous behaviour fosters prosocial behaviour, bonding, and affiliation between individuals (e.g., [55–57]; for a

meta-analysis, see [58]), we cannot exclude the possibility that the synchronous nature of the imagined joint actions might have had an additional effect on participants' IPD preference. A systematic comparison between non-synchronous and synchronous joint action scenarios is needed to determine to what extent, if at all, interpersonal synchrony plays a role in individuals' IPD preferences.

As already pointed out in the Introduction, we considered that, due to the time when this research was conducted, COVID-related concerns might naturally have an effect on participants' IPD preferences. Recent research has already documented that people preferred to keep farther apart from other individuals during the pandemic (e.g., [40, 41]). We predicted that the Perceived Infectability dimension of the PVD scale might positively predict preferred IPD as it provides an index of individuals' subjective beliefs about their susceptibility to catch an infectious disease from other people–a factor that should be particularly relevant during social interaction. Moreover, we considered people's momentary concern about COVID-19 as another potential predictor of preferred IPD. Our results were partially in line with these hypotheses: participants' overall IPD preference was modulated by participants' discomfort in scenarios with the potential for pathogen transmission (as measured by the dimension Germ Aversion of the PVD scale) and by their momentary concern about COVID-19. As such, participants who reported higher discomfort for potential pathogen contact and who were more aware of the COVID-19 context preferred to keep further away from a co-acting individual. The influence of Germ Aversion on preferred IPD is in keeping with the findings of Hromatko and colleagues observed during the early stages of the COVID-19 pandemic [51]. Similarly, Geers and Coello [50] found that individuals higher in Germ Aversion preferred to keep a greater distance from individuals who were not wearing a face mask. Perhaps surprisingly, the Perceived Infectability dimension turned out not to be a significant predictor of preferred IPD. Further research is required to examine the two dimensions (Germ Aversion and Perceived Infectability), in particular the extent to which they might play a role in naturalistic social interaction.

Finally, several limitations of the present research should be considered. First of all, we did not ask participants to engage in real-life interactions but rather to simply imagine these interactions (based on written scenario descriptions). To capture people's real-life preferences (and to verify the ecological validity of our experimental results), one should try to replicate this study using real-life scenarios [59]. It should be noted, however, that effects observed using the sort of IPD rating scale used here tend to align well with effects observed using more naturalistic methods [47]. A second, related concern pertains to the limited number of scenarios used in the present study. It would be worthwhile to add more scenarios and thereby widen the scope of the examined interactions. This was not done in the present study because it was the very first attempt to use this method for examining potential IPD differences between joint and parallel action (and thus should serve as a starting point for further investigations), and also because creating matching naturalistic joint/parallel scenario versions, that differ only minimally in regard to other factors, is challenging. For example, in the Exercise Class and Drumming Lesson scenarios in Experiment 2, the parallel versions featured movements performed at separate times. This was to avoid individuals imagining that their different movements were contributing to an overarching goal (performing for a teacher). This is not strictly temporally parallel like the situations described in the Storm Damage and Cafeteria scenarios. However, parallel action is not so commonly studied, and therefore readily defined, as joint action. We also reiterate that differences in preferred IPD were observed for almost all scenarios indicating that, at the very least, IPD preference is reduced for joint action compared to when similar behaviour is performed individually in temporal or spatial proximity to another. Another limitation concerns the participant sample, which only contained people living in two

Western European countries (UK and Germany). Thus, it remains unclear whether our results generalise to different cultural contexts, although we note that whilst the magnitude of preferred IPD can vary across cultures in general, the direction of effect for different social contexts (e.g., familiarity with the other) may potentially be similar [14].

In conclusion, the present work provides initial empirical evidence suggesting that the interpersonal distance people prefer to keep from others is influenced by the type of social interaction they are involved in, such that they prefer smaller interpersonal distances when acting jointly compared to in parallel with others. Further research is needed to validate and generalise these findings, as well as discover why they might occur.

## Supporting information

**S1 Table. Perceived vulnerability to disease scale items.** Item numbers for Perceived Infectability dimension in bold. Item numbers for Germ Aversion dimension in italics. Reverse-coded items are indicated with an asterisk.
(PDF)

**S2 Table. Multiple regression analysis for *diffIPD* (Experiment 1).**
(PDF)

**S3 Table. Multiple regression analysis for *diffIPD* (Experiment 2).**
(PDF)

## Acknowledgments

This work was supported by the European Centre for Advanced Studies (ECAS), Leibniz University Hannover, and the University of Stirling. The authors would like to thank Dr Magdalena Ietswaart and Dr Melanie Krüger for their feedback on project development, Daisy Martin for her assistance with piloting and data analysis, and Dr Catriona Scrivener for her feedback on a draft version of the manuscript.

## Author Contributions

**Conceptualization:** Laura Schmitz, Arran T. Reader.

**Data curation:** Laura Schmitz, Arran T. Reader.

**Formal analysis:** Laura Schmitz, Arran T. Reader.

**Investigation:** Laura Schmitz, Arran T. Reader.

**Methodology:** Laura Schmitz, Arran T. Reader.

**Project administration:** Laura Schmitz, Arran T. Reader.

**Validation:** Laura Schmitz, Arran T. Reader.

**Visualization:** Arran T. Reader.

**Writing – original draft:** Laura Schmitz, Arran T. Reader.

**Writing – review & editing:** Laura Schmitz, Arran T. Reader.

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
