## [Decision Letter · Decision Letter 0]

31 Jan 2023

PONE-D-22-31690Smaller preferred interpersonal distance for joint versus parallel actionPLOS ONE Dear Dr. Reader,

Thank you for submitting your manuscript to PLOS ONE. We have now received the feedback from two independent reviewers, which are both positive overall. As you will see, Reviewer 2 listed a few minor concerns which I urge you to address.

I particularly agree with Reviewer 2 regarding clarity of the approach to joint/parallel action and aims of your study. For example, in H1 you used moving in synchrony in the exercise scenario as a prime for joint action whereas in H2 you used moving in synchrony in an exercise scenario as a prime for parallel action. We tend to adjust our movements to synchronise those of others automatically and moving in synchrony is inherently enjoyable (e.g., mirroring). Thus, with the goal of enjoyment, I would argue that if done consciously, moving in synchrony is a joint action. I think further elaboration here and in regards to the novelty of your study as pointed out by Reviewer 2 would strengthen your work.

In addition to the rating scale comments of Reviewer 2; I am concerned that your visuals are not context sensitive enough. As shown in your Figure 1, participants have to mentally translate a distance from the left to right to a figure to their own personal space. The paper you cited that used a similar approach is quite old (1973). Other more recent studies used more interactive approaches, including the other citation from 2016, which showed differences between active (moving) and passive (still) contexts. Would you expect that an active task using a 3D visualisation to be more sensitive than your IPD rating scale (that has considerably low ecological validity)?

My final concern was not highlighted by the reviewers and is regarding your analyses. Why did you run so many separate analyses with different calculated dependent measures (e.g., meanIPD)? I understand that you have pre-registered your approach, however, I understand that this was not peer-reviewed. I also do not understand why you report an impact of Covid related concerns in the discussion when the null model without them was clearly fitting the data better.

Therefore, we invite you to submit a revised version of the manuscript that addresses all the points raised during this reviewing cycle.

Please see details below. 

We look forward to receiving your revised manuscript.

Kind regards,

Corinne Jola

Academic Editor

PLOS ONE

Journal Requirements:

Reviewers' comments:

Reviewer's Responses to Questions

**Comments to the Author**

1. Is the manuscript technically sound, and do the data support the conclusions?

Reviewer #1: Yes

Reviewer #2: Yes

2. Has the statistical analysis been performed appropriately and rigorously? 

Reviewer #1: Yes

Reviewer #2: Yes

3. Have the authors made all data underlying the findings in their manuscript fully available?

Reviewer #1: Yes

Reviewer #2: Yes

4. Is the manuscript presented in an intelligible fashion and written in standard English?

Reviewer #1: Yes

Reviewer #2: Yes

5. Review Comments to the Author

Reviewer #1: This study used surveys to study interpersonal distance preference during different type of social scenarios, namely joint actions where people are interacting towards a shared goal and parallel actions that may share a similar motor context but with no collaborative intent.

This paper is extremely well written. The hypotheses are clearly exposed, the methods are sound, and the authors are in line with open-science and reproductible science practices (pre-registration, preprint, power analysis, raincloud plots).

The authors predicted smaller preferred IDP for joint actions compared to parallel ones, and their empirical data clearly validate this prediction. Although this is not surprising per se, it is an important result to add to the field. It also validates online surveys as a good tool to inquire about socio-cognitive processes.

I congratulate the authors for doing such a good job and I thank them for making my job as a reviewer very easy.

Reviewer #2: The authors tested whether the nature of the social interaction (joint vs parallel) and concerns about pathogen infection affect preferred interpersonal distance (IPD). The results showed that participants would prefer shorter IPD to perform joint as compared to parallel actions. This difference was not affected by concerns about pathogens but the overall preferred IPD (averaged over the two types of interaction) was, with people aversive to germs preferring larger IPD. The methods are particularly sound, e.g., large sample size (N = 200), replication experiment to control for the potential methodological issues of the first experiment, and pre-registration. I only have one relatively major concern and several minor concerns:

- The original contribution of this study compared to previous studies is not sufficiently clear. Previous studies already showed an effect of collaboration vs. competitive tasks on preferred IPD. The additional contribution of studying joint vs. parallel actions should be developed as soon as in the introduction. Moreover, it has already been shown that Croat participants with germ aversion prefer to keep a larger IPD during the first months of the pandemic (Hromatko et al., 2021). More than two years later germ aversion has been found to have no direct effect on the French people’s IPD, but to modulate the effect of face masks on IPD, with people highly aversive to germs preferring confederates without face mask to be put further away (Geers & Coello, 2022). As for the current paper, only Germ Aversion (and not Perceived Vulnerability) had an effect. The author might cite these results in the introduction and discuss them in relation to theirs.

Minor:

- The example of joint vs. parallel actions in the introduction might not be the most relevant since it confounds the nature of the social interaction with the nature of the relation between the two confederates.

- The reliability of the Visual Analog Scale might already be presented in the Methods to justify their choice.

- Was the actual size of the rating scale on the participant’s screen controlled for? Position judgments depend on the physical length of the scale they have to be made on. For instance, the middle of a line is positioned towards the right vs. left as compared to the actual middle for short vs. long lines (McCourt & Jewell, 1999). So, performing the IPD rating on the screen of a smartphone vs. computer might induce some differences (and thus noise here) related to the size of the scale.

- The plot of the individual data is not very legible. The authors might consider decreasing the size of the dots.

- I would announce the objectives of Experiment 2 in the introduction. The issues of the scenarios in Experiment 1 personally disturbed me until reading they would be addressed in Experiment 2.

- Could the authors provide the corrections they made to the scenarios in Experiment 2?

- Another potential reason for the difference between the Strom Damage scenario and the other scenarios in Experiment 2, is that it requires manipulating a saw, i.e., a dangerous object that one generally avoids manipulating in the vicinity of someone.

6. PLOS authors have the option to publish the peer review history of their article (what does this mean?). If published, this will include your full peer review and any attached files.

Reviewer #1: **Yes: **Quentin Moreau

Reviewer #2: **Yes: **Laurie Geers

---

## [Decision Letter · Decision Letter 1]

18 Apr 2023

Smaller preferred interpersonal distance for joint versus parallel action

PONE-D-22-31690R1

Dear Dr. Reader,

We’re pleased to inform you that your manuscript has been judged scientifically suitable for publication and will be formally accepted for publication once it meets all outstanding technical requirements.

Kind regards,

Srebrenka Letina, Ph.D.

Academic Editor

PLOS ONE

Additional Editor Comments (optional):

I am happy to inform you that your last manuscript version is accepted for publication.

Reviewers' comments:

Reviewer's Responses to Questions

**Comments to the Author**

1. If the authors have adequately addressed your comments raised in a previous round of review and you feel that this manuscript is now acceptable for publication, you may indicate that here to bypass the “Comments to the Author” section, enter your conflict of interest statement in the “Confidential to Editor” section, and submit your "Accept" recommendation.

Reviewer #1: All comments have been addressed

Reviewer #2: (No Response)

2. Is the manuscript technically sound, and do the data support the conclusions?

Reviewer #1: Yes

Reviewer #2: Yes

3. Has the statistical analysis been performed appropriately and rigorously? 

Reviewer #1: Yes

Reviewer #2: Yes

4. Have the authors made all data underlying the findings in their manuscript fully available?

Reviewer #1: Yes

Reviewer #2: Yes

5. Is the manuscript presented in an intelligible fashion and written in standard English?

Reviewer #1: Yes

Reviewer #2: Yes

6. Review Comments to the Author

Reviewer #1: The paper is suitable for publication, the authors did a fine job with their rebuttal letter.

Reviewer #2: I still believe that the objectives of the studies could have been strengthen but otherwise the authors adressed all my comment.

7. PLOS authors have the option to publish the peer review history of their article (what does this mean?). If published, this will include your full peer review and any attached files.

Reviewer #1: **Yes: **Quentin Moreau

Reviewer #2: **Yes: **Laurie Geers

---

## [Editor Report · Acceptance letter]

24 Apr 2023

PONE-D-22-31690R1 

Smaller preferred interpersonal distance for joint versus parallel action 

Dear Dr. Reader:

I'm pleased to inform you that your manuscript has been deemed suitable for publication in PLOS ONE. Congratulations! Your manuscript is now with our production department. 

Kind regards, 

on behalf of

Dr. Srebrenka Letina 

Academic Editor

PLOS ONE